# The Therapeutic Potential of the Specific Intestinal Microbiome (SIM) Diet on Metabolic Diseases

**DOI:** 10.3390/biology13070498

**Published:** 2024-07-04

**Authors:** Natural H. S. Chu, Elaine Chow, Juliana C. N. Chan

**Affiliations:** 1Department of Medicine and Therapeutics, The Chinese University of Hong Kong, Prince of Wales Hospital, Hong Kong SAR, China; e.chow@cuhk.edu.hk (E.C.); jchan@cuhk.edu.hk (J.C.N.C.); 2Hong Kong Institute of Diabetes and Obesity, The Chinese University of Hong Kong, Prince of Wales Hospital, Hong Kong SAR, China; 3Li Ka Shing Institute of Health Sciences, The Chinese University of Hong Kong, Prince of Wales Hospital, Hong Kong SAR, China

**Keywords:** prebiotics, metabolic diseases, fermentable carbohydrates, specific intestinal microbiome, diabetes, NAFLD, obesity, metabolic cardiovascular diseases

## Abstract

**Simple Summary:**

Prebiotics, essential for humans and our gut microbiome, maintain the ecosystem’s homeostasis in a mutual relationship with the host and microbiome. As a gut microbiome modulator, poorly absorbed or indigestible carbohydrates interact with the gut microbiome, and its metabolites promote immune health. However, there is limited discussion of a habitual diet in metabolic diseases. Exploring the intricate crosstalk between dietary prebiotics and the specific intestinal microbiome (SIM) is intriguing to gain deeper insights into their therapeutic implications.

**Abstract:**

Exploring the intricate crosstalk between dietary prebiotics and the specific intestinal microbiome (SIM) is intriguing in explaining the mechanisms of current successful dietary interventions, including the Mediterranean diet and high-fiber diet. This knowledge forms a robust basis for developing a new natural food therapy. The SIM diet can be measured and evaluated to establish a reliable basis for the management of metabolic diseases, such as diabetes, metabolic (dysfunction)-associated fatty liver disease (MAFLD), obesity, and metabolic cardiovascular disease. This review aims to delve into the existing body of research to shed light on the promising developments of possible dietary prebiotics in this field and explore the implications for clinical practice. The exciting part is the crosstalk of diet, microbiota, and gut–organ interactions facilitated by producing short-chain fatty acids, bile acids, and subsequent metabolite production. These metabolic-related microorganisms include *Butyricicoccus*, *Akkermansia*, and *Phascolarctobacterium.* The SIM diet, rather than supplementation, holds the promise of significant health consequences via the prolonged reaction with the gut microbiome. Most importantly, the literature consistently reports no adverse effects, providing a strong foundation for the safety of this dietary therapy.

## 1. Introduction

A wise food selection is fundamental for the health and lifespan of individuals with metabolic diseases. The consequences of metabolic imbalance may be severe, and diet can provide the essential energy, macronutrients, and micronutrients for growth, cell differentiation, repair, and maintenance. The current dietetic guideline is not a rigid prescription and is affected by personal, cultural, and traditional preferences. Many recommendations have remained relatively consistent over time; they suggest that a balanced diet with nutrient-dense foods that provide vitamins, minerals, and other health-promoting components in reducing the risk of development of metabolic diseases, such as obesity, diabetes, cardiovascular diseases, and non-alcoholic fatty liver [1,2]. Different phenotypes of the human body express the pros and cons of dietary effects, such as anthropometric, biochemical parameters, and the gut microbiome. In recent years, there has been increasing interest in the role of the intestinal microbiome in the development and treatment of metabolic diseases. The gut microbiota, a fascinating and complex ecosystem within our bodies, plays a significant role in metabolic diseases. However, nutrient-dense foods, known as functional foods, need to be better defined, mostly suggested in categories of vegetables, fruits, whole grains, seafood, legumes, unsalted nuts and seeds, and fat-free and low-fat dairy products. In current dietetic knowledge, some foods reshape the gut microbiota and produce vital health-related elements; dietary fiber is the most discussed topic. However, apart from the dietary fiber, other short or medium-chain carbohydrates, such as oligosaccharides and inulin [3,4], also induce the specific intestinal microbiome (SIM) and their metabolites, such as secondary bile acids and short-chain fatty acids (SCFAs), which are important in the etiopathophysiology of metabolic diseases [5,6]. They communicate with the gut–organ axis, hormones, metabolic pathways, and immune cells [3,5,7,8]. Bioactive metabolites that originate from the gut microbiota are known to be produced through consuming natural food. Aromatic amino acids such as tryptophan, primary bile acids, and others are the potential substances that link the microbiota to the host’s physiology, a so-called mutualism. Furthermore, many studies have confirmed the beneficial effects of the increase in SCFAs, the reduction of inflammation and endotoxemia, and the involvement of gut microbiota assistance in a range of bodily functions. These include the protection of pathogens and the regulation of immune functions [8,9,10]. The effects of macronutrient metabolism by the gut microbiome on human health have been extensively studied, but research on the impact of short and medium-chain carbohydrates on the gut microbiome in metabolic diseases remains limited [3]. In this review, we gather the particular carbohydrate polymers that interact with the SIM and affect host metabolism in a diet-specific manner to treat or prevent metabolic diseases. First, we need to understand the mechanism of the SIM, which works in the pathophysiology of the prevention of metabolic diseases. Secondly, there is evidence of prebiotics and potential prebiotics stimulating the SIM and producing SCFAs and other microbial metabolites. Finally, evidence of dietary effects rather than the supplementation of prebiotics is shown. Understanding these data empowers people to take control of their metabolic health through their nutritional choices. 

## 2. SIM and Metabolic Diseases

The complexities of the gut microbiota and their influence on metabolic health require a multidimensional perspective, encompassing factors such as dietary composition, individual variations in microbiome composition, and their collective impact on metabolic function. Dysbiosis, a disruption of the gut microbiota, is involved in different metabolic progressions. However, gut microbiota profiles lack uniformity due to environmental confounders, study design, medication, health conditions, and ethnicity. In 575 individuals with diabetes mellitus and 840 healthy controls, there are significant correlations in specific gut microbiomes. Individuals with diabetes displayed a lower abundance of *Bifidobacterium* and higher levels of *Lactobacillus* spp. compared with non-diabetics [11]. The obesity-related microbiome included the genus of *Megamonas* and *Escherichia–Shigella* as biomarkers of obesity [12]. Abnormalities of gut microbiota composition in patients with non-alcoholic fatty liver disease (NAFLD) have been highlighted in a meta-analysis showing an increased abundance of *Escherichia*, *Prevotella*, and *Streptococcus* and a decreased abundance of *Coprococcus*, *Faecalibacterium*, and *Ruminococcus* [13]. A higher abundance of *Clostridium sensu stricto*, *Desulfovibrio*, *Parabacteroides*, and *Streptococcus* and a lower abundance of Ruminococcaceae, *Roseburia*, and *Faecalibacterium* spp. are found in individuals with hypertension [14]. These meta-analyses overlap with several microbial signatures in metabolic diseases. Nevertheless, the studies mentioned were not able to investigate the dietary factors, highlighting the urgent need for further research in this area.

The gut is a complex ecosystem that contains a diverse array of bacteria, including commensal, harmful, and conditional bacteria. Commensal bacteria act on the host’s immune system to induce protective responses that prevent colonization and invasion by pathogens. Multiple studies have established associations between human gut bacteria and host physiology, especially the genera of *Bifidobacterium*, *Akkermansia*, and *Lactobacillus*. In the previous review, we summarized the characteristics of gut microbiota in newly diagnosed diabetes, and these microbiota patterns can be restored with antidiabetic treatments provided by Western and Chinese medicine [15]. Considering the significant impact of drugs on reshaping the gut microbiome, the administration of probiotics stands out as a direct and effective method for microbiome manipulation. In a meta-analysis of the effects of probiotics on metabolic disease-related variables, the intake of probiotics resulted in consistent improvement in anthropometric and biochemical parameters in individuals with metabolic diseases, although some had a minor effect [16,17]. The possible cause depended on the survival conditions for the growth of probiotics in those patients with metabolic diseases, such as the food being fed to the gut microbiota. The human gut microbiome is a complex and dynamic ecosystem in which different strains of microorganisms compete for different resources to effectively colonize it [18]. Our selection of foods is not only needed for basal metabolism, body development, and essential vitamins but also for feeding our gut microbiota, as food provides “prebiotics”. Prebiotics are an element of food that promotes the growth of beneficial microorganisms in the intestines, and the composition of the microbiota is predominated by carbohydrate polymers [19]. The combination of prebiotics and probiotics showed a better improvement of glycemic variables than probiotics alone, as summarized in a systematic review of randomized controlled trials [20]. However, based on the current definition of prebiotics and accumulating evidence, some food ingredients exhibit the prebiotic effect but have yet to be classified as prebiotics. Therefore, this review summarizes the evidence of natural food ingredients that stimulate and maintain SIM growth for the prevention or treatment of metabolic diseases.

## 3. Crosstalk of Gut Microbiome and Its Metabolites

Undoubtedly, the gut microbiota assumes a pivotal role in maintaining the gut’s integrity, acting as a robust defense. Equally important is its role in the production of a diverse range of bioactive molecules, which are instrumental in interconnecting glycolysis, the tricarboxylic acid/Krebs cycle, oxidative phosphorylation, and macronutrient metabolism. In the molecular context, these bioactive metabolites derived from gut microbiota serve as crucial signaling molecules to various cell types, thereby facilitating hormone secretion [21]. A current animal study has uncovered the unique function of *Butyricimonas virosa*, which not only belonged to an SCFAs-producing bacteria but also exhibited a novel function in glucose regulation that was linked to the upregulation of GLP-1 receptors rather than its production of SCFAs [22]. Another recent study unveiled a lipid from *Akkermansia muciniphila*’s cell membrane that mimics the immunomodulatory activity of a15:0-i15:0 PE, which has a highly restricted structure–activity relationship [23]. This underscores the profound influence of gut microbiota on miRNA expression and the modulation of the host’s immune homeostasis [24,25]. For instance, microbiota activities have been shown to downregulate the expression of miR-10a in dendritic cells through the toll-like receptors TLR–TLR ligand TLR4, TLR5, TLR9, and nucleotide-binding oligomerization domain-containing protein 2 (NOD2) interaction via the myeloid differentiation factor 88-dependent pathway. This downregulation effectively reduces NF-κB and p38/Jun N-terminal kinase (JNK) activation, thereby potentially mitigating intestinal inflammation as observed in an animal study [26]. A study involving a blinded, randomized, cross-over dietary intervention on healthy individuals further solidified these findings [10]. The results showed that a significant increase in colonic and peripheral blood SCFAs, blood total B cells, naïve B cells, and mucosal-associated invariant T cells was observed in the high-SCFAs diet group compared to the low-SCFAs diet group over 21 days [10]. 

SCFAs, the most common bioactive metabolites produced by the intestinal gut microbiota, are fascinating due to their diverse effects on various physiological processes and the novel mechanisms they employ. For instance, specific SCFAs, such as acetate, butyrate, and propionate, have been found to activate G protein-coupled receptors (GPR41, GPR43, and GPR109a) and inhibit histone deacetylase (HDAC), mechanisms that are intriguing and warrant further exploration [27]. 

Firstly, the effects of SCFAs on metabolism and adipocyte lipolysis are multifaceted. They are involved in the activation of the activity of GPR43 [28,29] present in pancreatic islet α and β cells [30,31] and the secretion of the hormone leptin in adipocytes [32]. This can potentially lead to a reduction in insulin resistance. SCFAs also play a crucial role in regulating several leukocyte functions, including the production of pro-inflammatory cytokines such as TNF-α and IL-1β, anti-inflammatory cytokines, such as IL-10 and TGFβ, or glucocorticoids, eicosanoids, and chemokines (e.g., MCP-1 and CINC-2) [27,33]. An increase in SCFAs can reduce systemic lipopolysaccharide (LPS) endotoxemia inflammation and activation of toll-like receptor-4 and reverse the production of oxidative stress [34,35] (reactive oxygen species) and NF-kB transactivation. This results in an increase of carnitine palmitoyltransferase 1 (CPT1), peroxisome proliferator-activated receptor alpha (PPARα) target gene, and a decrease in IL-8 mRNA transcription that may improve apolipoprotein A-I formation of dysfunctional HDL particles, which is consequently linked to the reduction of cardiometabolic inflammation [36]. Additionally, the observations are in line with the studies by Brown et al. and Ørgaard A. et al., which demonstrated significant inhibition of lipolysis in primary human fat cells and stimulation of glucagon-like peptide-1 (GLP-1) [37] and somatostatin hormone secretion [38]. SCFAs were shown to decrease the serum concentration of ghrelin (appetite hormone) and stimulate the production of GLP-1 and Peptide YY by activating GPR43 and GPR41 [39]. This involved the mechanisms in the gut–brain neural circuit; an animal study observed that butyrate administration suppressed the activity of orexigenic neurons and decreased neuronal activity in the brainstem. Therefore, this reduced the appetite and activated brown adipose tissue by utilizing plasma triglyceride-derived fatty acids [40]. More specifically, in one of the SCFAs, acetate, a study used labeled carbohydrate and positron emission tomography (PET)-scanning to investigate the SCFA and its effect on the host’s appetite and showed that intraperitoneal acetate was fermented by the gut microbiome and taken by the brain, and this resulted in appetite suppression and hypothalamic neuronal activation patterning [41]. These interactions provide the mechanical benefit of reducing systemic inflammation, adiposity, insulin resistance, and cardiometabolic and diabetes risk. Secondly, SCFAs are natural inhibitors of HDACs, which play an essential role in chromosome structure modification and gene expression regulation via the dissociation of DNA and the relaxation of nucleosome structure. HDACs upregulate genes associated with fatty acid uptake and oxidation, electron transport, or oxidative phosphorylation accompanied by fatty acid-induced myocardial lipid accumulation and elevated triglyceride levels shown in many animal studies [42,43]. SCFA-mediated HDAC inhibition promotes chromatin acetylation and reduces cardiovascular complications, offering a promising avenue for future research and potential therapeutic interventions.

Another important microbiota-medicated metabolite is secondary bile acids. Primary bile acids, such as cholic acid and chenodeoxycholic acid, play critical roles in cholesterol metabolism and lipid digestion, and bile acids are reabsorbed via enterohepatic circulation. Primary bile acids will be transformed by gut microbiota after the deamination of conjugated bile acids, mainly from the genera *Enterobacter*, *Clostridium*, and *Enterococcus* [44], to secondary bile acids, such as deoxycholic acid and lithocholic acid or ursodeoxycholic acid (UDCA) in humans, specifically in the colon which affects the diverse metabolic pathways. For instance, a water-soluble bile acid, tauroursodeoxycholic acid (TUDCA), is produced by the conjugation of UDCA with taurine in the body. This compound has shown a 30% improvement in hepatic and muscle insulin sensitivity in people with obesity compared to the placebo [45]. Another promising interaction of conjugated secondary bile acid is glycodeoxycholic acid (GDCA), which forms with deoxycholic acid and another amino acid, glycine, in the liver. GDCA was correlated with reduced insulin resistance via communication with the gut microbiome and interleukin-22 secretion [46]. 

It is notable that an increase in a secondary bile acid stimulates the farnesoid X receptor (FXR) expressed in the liver and intestine and leads to the release of fibroblast growth factor (FGF) 19 in humans or FGF15 in mice, which improves insulin sensitivity and hepatic lipid metabolism [47]. An interesting observation was made in Roux-en-Y gastric bypass (RYGB) patients who showed increased postprandial bile acid and FGF19 response compared with obese controls [48]. This suggests a potential link between the metabolic effects of secondary bile acids and weight loss. Secondary bile acids can also activate the thiol guanosine receptor-5 (TGR5), boost muscle energy consumption, stimulate the secretion of GLP-1 by endocrinal L cells, and suppress the dysbiosis of pathogenic bacteria [49]. Furthermore, the interaction between GLP-1 and natural gut intraepithelial T lymphocytes around the small intestine regulates an overabundance of dietary fat and simple sugar systemic metabolism [50]. This microbial biotransformation included bacteria belonging to the genera *Bacteroides*, *Clostridium*, *Eubacterium*, *Lactobacillus*, and *Escherichia* [51]. A multicenter, double-blind, randomized, placebo-controlled, phase 3a study on 283 patients with non-alcoholic steatohepatitis (NASH), with and without type 2 diabetes, demonstrated that 72 weeks of treatment with semisynthetic bile acid derivative obeticholic acid led to significant improvement in the NASH activity score [52]. In addition, alteration of secondary bile acids metabolism inhibited bile acid synthesis in the liver [53], the involvement of deamination, and regulated ammonia hepatic removal, which may contribute to a reduction in hepatic ammonia accumulation underlying steatosis [54]. Abnormality of bile acids influences vasodilation and transcription of vasoactive molecules via large conductance calcium-activated potassium channels, causing the risk of cardiovascular disease [55]. Therefore, altering secondary bile acids can be a potential therapeutic intervention for metabolic-associated fatty liver disease (MAFLD) and other metabolic diseases. Furthermore, Figure 1 illustrates the mechanism of dietary prebiotics involving the intestinal crosstalk of the gut microbiome and its metabolites in metabolic diseases.

## 4. Current Evidence of Food for the Simulation of the SIM

In the past, fiber was shown to be the most essential ingredient for allocating intestinal microbiota. However, not all ingredients classified by the definition of fiber influence the metabolic health or ecosystem in the gut microbiome [56]. For example, a meta-analysis by He and colleagues did not find associations between beta-glucans and HbA1c, fasting glucose, and fasting insulin in overweight individuals or those with T2D or hyperlipidemia. However, the study did find associations between oats (which are high in oligosaccharides) and the above variables [57]. This research underscores the need for a nuanced understanding of food ingredients and their effects on gut microbiota and metabolic health. Food elements that cannot be digested or absorbed by the host will leave bacteria as food, defined as “prebiotics” [58]. The inter-individuals’ responses to prebiotics and gut microbiomes require personalized nutrition strategies. For example, a human study investigated the relationship between the administration of *Bifidobacterium* strains and galato-oligosaccharides (GOS) in obese adults; they did not find the synergism as a symbiotic effect [59]. Understanding the specific food ingredients, especially in carbohydrate polymers, can manipulate the target microbiota for reversing dysbiosis. This understanding is not just theoretical but has practical implications in managing metabolic diseases. Some short-chain carbohydrate polymers, such as GOS and fructose-oligosaccharides (FOS), overlap with fiber. Apart from common prebiotics, resistant starch, polyols, and other undigested disaccharides and monosaccharides were partly digested or not fully absorbed by the small intestine and then passed through the large intestine for the fermentation process by gut microbiota [60]. However, insulin-resistant individuals exhibit increased fecal monosaccharides, suggesting that the dysbiosis in those individuals reduced the microbial carbohydrate metabolism [61]. In this review, we suggest that some poorly absorbed or indigestible carbohydrates promote the SIM, which is beneficial for carbohydrate metabolism to manage metabolic diseases. Figure 2 shows the classification of fermentable carbohydrates.

## 5. Soluble Non-Starch Carbohydrates (NSPs)

Soluble non-starch polysaccharides, notably pectin and psyllium, offer promising health benefits due to their prebiotic properties. The role of NSP structure and function in health advantages such as antioxidants, anti-inflammatory, anti-diabetic, lipid-lowering, and immunomodulatory effects have been well discussed [62,63,64]. In animal studies, pectin has shown potential in reducing fasting glucose levels [65], liver steatosis [66], and serum lipid profiles [67,68]. Pectin specifically favored the growth of *Bacteroides* and also increased the species *Lachnospira eligens* and *Faecalibacterium prausnitzii* [69]. However, the effect of pectin is less compelling in human studies. Another promising viscous fiber is psyllium, which has been shown to be more effective at decreasing serum lipopolysaccharides, serum triglycerides, and liver cholesterol than the anti-obesity drug (orlistat) by increasing the relative abundance of *Roseburia*, *Bacteroides*, *Faecalibacterium*, and *Coprobacillus* [70]. Due to its unique chemical structure, psyllium forms a viscous gel that binds to bile acids in the gut and eliminates bile acids via the stools, thus reducing blood cholesterol concentrations. Psyllium husk did not significantly reduce the fasting and postprandial glucose in animal studies, but significant results were shown in human studies [71,72,73]. More importantly, the significant cholesterol-lowering effects of psyllium were highly discussed. However, none of the studies investigated the effect of psyllium on gut microbiota in individuals with metabolic diseases. 

## 6. Oligosaccharides

Oligosaccharides have been widely discussed and commonly used as prebiotics. As discussed in the previous review [3], GOS and fructo-oligosaccharides (FOS) have been widely used to treat metabolic diseases. Dietary GOS contain raffinose family oligosaccharides and isomalto oligosaccharides (IMOs) that were correlated with lower body fat and higher insulin sensitivity in individuals with impaired glucose tolerance [74]. In another randomized controlled trial involving 29 patients with diabetes, GOS were associated with an increased level of gut microbiota belonging to the bacterial family Veillonellaceae (genus *Akkermansia*), which correlated inversely with glucose response [75]. Another type of commonly known oligosaccharide is Mannan, which is converted by Galactomannan polysaccharides (soluble fiber) during heating, involvement of hydration and swelling, disruption of hydrogen bonds, gel formation, and increased viscosity in industrial processes [76]. A recent in vitro fermentation study showed that Mannan-oligosaccharides can stimulate the growth of *Barnesiella*, *Odoribacter*, *Coprococcus*, and *Butyricicoccus* and increase SCFA production in a dose-dependent manner [77]. FOS are used separately as a prebiotic; they reduce postprandial blood glucose response and increase GLP-1 and PYY secretion [78,79]. In a meta-analysis, FOS supplementation was suggested to increase the number of *Bifidobacterium* spp. However, it depends on the duration and dose manner [80]. 

## 7. Resistant Starch

Resistant starch (RS) has been suggested as a prebiotic used for the utility of resistant starch in alleviating the burden of metabolic diseases by reconstructing gut microbiota. RS is classified into five types (RS1-5): physically inaccessible, enzyme inaccessible, retrogradation, chemically modified to complexed with lipid, and RS2-4, which can be degraded by *Ruminococcus bromii* or *Bifidobacterium adolescentis*. This degradation process can facilitate the regulation of intestinal motility, a key factor in maintaining gut health and potentially influencing metabolic health, by promoting the secretion of PYY [81]. Examples from animal studies suggested that RS increases *Akkermansia* and butyric acid, which are linked to metabolic health, as mentioned before [82,83]. In a human study, RS administration produced similar findings to those in animal studies; it reshaped the relative abundance of *Akkermansia*, *Ruminococcus*, *Victivallis*, and *Comamonas* compared to the baseline [84]. The protective effect of starch from Pueraria lobata to relieve non-alcoholic fatty liver (NAFLD)-associated gut dysbiosis was linked to increased quantities of *Lactobacillus*, *Bifidobacterium*, and *Turicibacter* and decreasing *Desulfovibrio* [85]. In a double-blinded, controlled study, RS4 significantly and clinically reduced blood glucose and insulin excursions compared with refined wheat flour [86]. In another study comparing different types of RS, large amounts of short-chain fatty acids with RS5 produced more butyric acid, and RS3 produced more lactic acid. Moreover, RS5 increased the relative abundance of *Bifidobacterium*, *Dialister*, *Collinsella*, *Romboutsia*, and *Megamonas* [87] significantly.

## 8. Inulin

Inulin has shown the potential benefits of managing diabetes mellitus and obesity, [4] but notably, the effects of inulin can vary among individuals. For instance, in a randomized, double-blinded study of a dietary intervention with 4 g/day of inulin after 3 months, individuals with diabetes showed an increased abundance of *Faecalibacterium prausnitzii* and *Akkermansia muciniphila* accompanied by decreases in serum glucose and HbA1c [88]. This outcome suggests a potential benefit of inulin in diabetes management. Furthermore, inulin administration promoted the expansion of *Bifidobacterium*, *Phascolarctobacterium*, and *Blautia* and reduced the opportunistic pathogens, such as *Acinetobacter* and *Corynebacterium*, in the inulin-treated rats. This study showed that inulin regulated the fecal metabolites of indole-3-acetic acid and kynurenic acid, whereas there were reduced levels of kynurenine and 5-hydoxyindoleacetic acid [89]. These findings highlight the potential of inulin in modulating gut microbiota and metabolic health. However, in a randomized, controlled, cross-over study comparing the consumption of refined grain, inulin, and wheat germ (15 g/d), the consumption of inulin caused no reduction in glucose AUC. Wheat germ (which contains GOS) caused the greatest reduction in the greatest glucose AUC among the three interventions [90]. This variability in outcomes underscores the need for further research and individualized approaches. Another study indicated a non-significant improvement in insulin levels and Homeostatic Model Assessment (HOMA) indices in individuals with obesity, but it found significantly decreased total cholesterol and LDL cholesterol after inulin supplementation [91]. Inulin also significantly decreased cannabinoid receptor-1 and Patatin-like phospholipase-3 gene expressions in the liver, which are associated with total NAFLD activity scores [92]. 

## 9. Disaccharides and Monosaccharides

Disaccharides or monosaccharides may not be absorbed because of a deficiency in enzymes that digest lactose and fructose, which affect the gut microbiome. Lactose comes from dairy products. The prevalence of lactose malabsorption ranges from 51% to 89% in the Asian population [93]. The role of dairy products in human health has been extensively studied [94,95]; daily consumption of milk has been associated with a lower risk of cardiovascular diseases, colorectal cancer, metabolic syndrome, obesity, and osteoporosis, as observed in 41 meta-analyses [95]. Some studies have suggested that low-fat milk and yogurt are associated with a lower risk of diabetes [96]. However, mechanistic studies are limited to the investigation of the effect of lactose on metabolic diseases. The current evidence suggests that the hypoglycemic effects of lactose are based on the increased protein intake from dairy products, which slow down sugar absorption [97]. A study in Finland investigated the postprandial blood glucose and insulin responses to liquid test meals in random order containing 40 g carbohydrates from milk, lactose, glucose, or fructose. Equal amounts of energy were compared in 10 patients with type 2 diabetes, and milk and lactose-based meals led to a similar glucose response, but there was a lower insulin response in milk than with the lactose meal [98]. However, few limited studies have investigated lactose and the gut microbiome. In an in vitro experiment investigating microbiota from healthy volunteers, lactose was found to decrease the relative abundance of Bacteroidaceae and increase lactic acid bacteria, specifically Lactobacillaceae, Enterococcaceae, and Streptococcaceae, and the health-promoting bacteria *Bifidobacterium* with increased total SCFAs, specifically acetate. Lactose also enhances the prevalence of the β-galactosidase gene [99], which expresses an essential glycoside hydrolase enzyme in the human body. Furthermore, lactulose, a non-digestible sugar derived from lactose, is also considered a prebiotic, increases the abundance of the SCFA-producing bacteria *Lactobacillus* and *Bifidobacterium*, and suppresses the potentially pathogenic *Escherichia coli* [100]. However, lactulose belongs to artificial sugars, and we discussed it as a prebiotic in the previous review [100]. 

High fructose consumption in animal studies causes hepatic and extrahepatic insulin resistance, affecting metabolic diseases such as obesity, diabetes, MAFLD, and high blood pressure [101,102]. Fructose ingestion increased the relative abundance of *Bacteroides fragilis* in mice, correlated with decreased secondary bile acid production [103]. However, the evidence is less compelling in humans and larger animals [104,105,106], diverging across evolution because the liver and kidneys are the only gluconeogenic organs in humans; the small intestine is not included as it does not express glucose-6-phosphatase (G-6-Pase) [107]. This critical issue is important in translating experimental evidence from mice to humans. Moore et al. conducted an oral glucose tolerance test (OGTT) of 75 g glucose with or without 7.5 g fructose on healthy controls and patients with type 2 diabetes. Even when the additional sugar dose increased by 10%, plasma glucose was reduced by 19% and 14% in healthy individuals and those with diabetes, respectively [108]. Of note, patients with diabetes experienced a reduction in plasma insulin (21%) when fructose was added [108]. Growing evidence shows that consumption of sweetened beverages (either sucrose or a high fructose corn syrup) is associated with increased body weight and postprandial triglycerides that increase the occurrence of metabolic and cardiovascular disorders [109,110]. This condition occurs because fructose is not absorbed from the gut as efficiently as the monosaccharide alone due to the low GLUT5 and GLUT2 activities in response to fructose [111]. However, we did not exclude the consideration that excessive fructose may increase insulin resistance through the co-absorption of glucose during digestion processes in humans. This condition can be observed in a study of healthy men who were administered a high dose of fructose (200 g/d) added to their usual meal, thus increasing their ambulatory blood pressure, triglycerides, fasting insulin, and HOMA indices. It also decreased HDL and resulted in the de novo development of metabolic syndrome in 25% of them [112]. Therefore, there is no unequivocal evidence that free fructose intake from natural foods is directly related to adverse metabolic effects and unabsorbed fructose in the gut increased the SCFA-producing genera *Anaerostipes Coprococcus*, *Ruminococcus*, and *Erysipelatoclostridium* in human studies [113,114] and the most abundant butyrate producer in the human gut is *Faecalibacterium prausnitzii*, which also ferments fructose [115]. These findings suggest that fructose is a two-edged sword depending on dose and nutrient co-absorption. 

## 10. Polyols

Polyols, also known as sugar alcohols, are compounds being investigated for their interactions with gut microbiota. Two common polyols, mannitol and sorbitol, are abundant in mushrooms and several fruits and vegetables. It is worth noting that polyols are also used to produce artificial sweeteners, such as lactitol and erythritol [116]. In a recent study, white button or portobello mushrooms were fed to animals for 15 weeks. Both mushrooms, being high in mannitol, significantly increased Verrucomicrobia and reduced Cyanobacteria [117]. More studies have explored the effects of mushroom polysaccharides (glycan and pectin) on gut microbiota. In vitro simulation experiments have demonstrated that mushroom polysaccharides promote the proliferation of beneficial bacteria, including *Bacteroides* and *Phascolarctobacterium* [117]. This reaffirms the prebiotic effect of mushrooms and contributes to a sense of their potential health benefits. Moreover, the polysaccharides inhibited the growth of unfavorable bacteria, such as *Escherichia–Shigella* [117], and had similar results to the intake of GOS in the previous session. While the effect of carbohydrate polymers from mushrooms on gut microbiota is still debatable, their prebiotic effect is undeniable.

## 11. Evidence for Prebiotics or Potential Prebiotics and Metabolic Diseases

Based on this evidence, the quality of carbohydrates, rather than the quantity, relieves metabolic diseases by regulating the gut ecosystem. This finding brings hope as it suggests that increasing the carbohydrate quality can minimize the risk of developing diabetes and other metabolic diseases [118]. For instance, a randomized cross-over study conducted on 50 individuals with high risks of metabolic syndrome revealed that a whole grain diet (high in oligosaccharides) led to a decrease in body weight, interleukin (IL)-6, and C-reactive protein after 8 weeks of dietary intervention, particularly with the intake of rye [119], which is rich in fructans. These positive outcomes bring a beacon of hope for health professionals, nutritionists, researchers, and individuals interested in metabolic diseases.

### 11.1. Obesity

In a meta-analysis of the use of psyllium, it is effective in reducing body weight and waist circumference [120]. Anti-obesity effects and inhibition effects of pathogenic microbiota using the treatment of oligosaccharides were observed in genetically and diet-induced obese animals, resulting in decreasing the inflammatory markers and reducing adiposity [121]. Similar results were also consistent in human studies, which increased the secretion of satiety hormones, decreased hepatic de novo lipogenesis, and eventually decreased food intake and fat-mass development [122]. An example of this is when individuals with diabetes add soluble fiber guar gum to their regular diet, which leads to improvements in waist circumference, HbA1c, 24 h urinary albumin excretion, and serum trans-fatty acids compared to their baseline levels. This also results in greater weight reduction compared to a control group [123]. In a randomized cross-over trial in healthy controls, resistant starch administration in controlled diets reduced the visceral and subcutaneous fat areas, LDL, and increased the production of GLP-1 and acetate [84]. In a placebo-controlled study in younger ages, consumption of inulin was found to reduce body weight, percent of body fat, and trunk fat compared to the baseline, but results were similar to those of the dietary fiber group [124]. Inulin increases the abundance of the genus *Bifidobacterium* and decreases *Bacteroides vulgatus*. In obese individuals, the pre-intervention levels of *Anaerostipes*, *Akkermansia*, and *Butyricicoccus* drive the decrease of BMI in response to inulin [125,126]. A meta-analysis and systematic review of cohort studies showed that the risk of overweight and obesity decreased by 7–25% due to increased dairy products [126]. 

### 11.2. NAFLD

In animal studies, fructo-oligosaccharides were administered to investigate the prebiotic effects on the hepatic manifestation of metabolic syndrome. The results were significant, showing a reduction of hepatic steatosis and liver inflammation [127,128]. This was accompanied by an increase in fecal short-chain fatty acid levels. This increase was linked to the change in gut microbiota, specifically the increase in beneficial bacteria such as Lactobacillales and *Clostridium* [127]. This change in gut microbiota is believed to play a crucial role in the observed reduction of hepatic steatosis and liver inflammation. Similarly, in a study of individuals with NAFLD, resistant starch administration led to a reduction in intrahepatic triglyceride content after adjustments of weight loss and the abundance of *Bacteroides stercoris* [129]. In a case-control study, fructose from fruits and vegetables was found to be inversely proportional to the odds of NAFLD in the Iranian population [130]. However, a high fructose diet (classic 60% caloric intake) causes impaired glucose tolerance and insulin resistance due to co-absorption with glucose, leading to increased sucrose metabolism [131]. Although the adverse effects of fructose ingestion in mice are stronger than in large animals or humans, it is crucial for individuals with NAFLD to be aware of their dietary choices and not exceed a daily intake of 50 g of fructose [3,132]. 

### 11.3. Diabetes

The evidence from 35 randomized, controlled clinical studies indicates that psyllium can significantly reduce fasting blood glucose and HbA1c levels [133]. Furthermore, it suggests that psyllium has the added benefit of boosting antioxidant activities and triggering signaling pathways that contribute to overall health. Psyllium effectively elevated the activities of superoxide dismutase (SOD), catalase (CAT), and glutathione peroxidase (GPX), which are entire defense strategies of antioxidants to inhibit hypoxia/reoxygenation-induced reactive oxygen species production and significantly enhanced the activation of the Akt/Nrf2/HO-1 signaling pathway [134]. Self-perceived lactose-intolerant respondents had reported a substantially higher rate of diabetes and hypertension that is linked to low intake of lactose suggested in these populations [135]. A current observational study indicated that higher consumption of GOS was negatively correlated with body fat and positively correlated with insulin sensitivity in prediabetes independent of physical activity, macronutrients, and fiber intake [74]. In a randomized, double-blind, placebo-controlled clinical trial, elderly patients with type 2 diabetes consumed milk powder co-supplemented with inulin and resistant dextrin and had a greater reduction of blood pressure and an increase in postprandial insulin and β-cell function index compared to placebo for 12 weeks [136]. 

### 11.4. Metabolic Cardiovascular Diseases

Plenty of randomized clinical studies and meta-analyses have evaluated the effect of psyllium on lipid levels and showed a significant improvement in systolic blood pressure [137]. Similar to psyllium, another soluble NSP, arabinoxylan consumption reduced fasting glucose, triglycerides, and apolipoprotein A-1 compared to placebo [138]. In a double-blind, randomized, placebo-controlled, cross-over study on overweight individuals with metabolic syndromes, GOS administration for 12 weeks decreased circulating cholesterol, triacylglycerols (TAGs), and total to HDL cholesterol ratios [139]. In another similar study, in which 10 g/day inulin treatment was given for three weeks in healthy individuals, inulin administration also resulted in decreased blood TAGs and liver lipogenesis [140]. In three large prospective cohorts in the US, a consistent result with a negative association between fructose intake and incident hypertension was seen [106]. Higher dairy products also showed negative associations with hypertension, either high or low-fat dairy [94], which is rich in another disaccharide (lactose). 

While many studies have focused on the effects of individual dietary components, the potential synergistic effects of prebiotics in the gut microbiome remain a promising yet unexplored area. This gap in our understanding presents an exciting opportunity for further research, potentially uncovering new strategies for improving health outcomes.

## 12. Dietary Effect Is More Promising than Prebiotics Administration

Based on the above studies, prebiotics and potential prebiotics address metabolic diseases. However, current evidence for prebiotic supplementation remains controversial. For example, adding 20 g/d inulin administration to an already high-fiber diet did not provide additional benefits on various physiological parameters, such as lipid profiles and body weight [141]. Other studies also had similar results on supplementation with inulin and/or FOS, which had no significant independent effects on fasting glycemia and insulinemia, HOMA-IR index, or lipid profile [142,143,144,145,146]. In a randomized controlled trial in individuals with type 2 diabetes, who were randomized to a prebiotic (galacto-oligosaccharide mixture) or placebo (maltodextrin) supplement for 12 weeks, GOS supplementation had no significant effects on clinical outcomes or bacterial abundance compared with a placebo [75]. This result may be linked to natural prebiotics rather than supplementation in altering regional gut transit time [147]. It is a direct way of preventing the over-absorption of macro/micronutrients and is secondary to systemic chronic inflammation and immune response [148]. Thus, diets containing potential prebiotics may maximize the regional gut transit time, thus increasing the reaction rate of gut microbiota [149]. An example of a feeding study measuring gastric emptying time in healthy controls who consumed liquid and solid breakfasts showed that both meals contained the same energy and fiber. Still, the solid meal had a longer gastric emptying time than the liquid meal, and individuals also felt less hungry on a solid diet [150]. However, limited studies have compared supplementation with prebiotics and dietary prebiotics in individuals with metabolic diseases. The importance of a personalized diet is geographic and cultural [151]. Due to species’ structure variance across geographies, examining food ingredients and microbiota in the local region is essential [152]. 

In summary, different types of prebiotics have been found to have specific effects on metabolic parameters. Soluble NSPs and inulin, for example, have been correlated with improved lipid profiles, weight loss, and blood pressure. GOS, on the other hand, have been associated with insulin sensitivity and weight loss. Resistant starch and fructans have shown a relationship with hepatic parameters, such as ALT and AST [153]. It is also worth noting that other short-chain carbohydrates interact highly with the gut microbiome. Table 1 contains examples of natural foods with high fermentable carbohydrate content and shows how they can impact the microbiome [3]. Including more of these natural prebiotics in our regular diet could be a new dietary approach to help manage metabolic diseases. 

## 13. Comparison between Current Dietary Therapies on Metabolic Diseases

Specific dietary modifications improve the food quality or change macronutrient distribution, showing beneficial effects on metabolic syndrome conditions and individual parameters. These specific nutritional modifications include the Mediterranean, plant-based, and dietary approaches to stop hypertension (DASH) diets as the paradigm for metabolic syndrome prevention and treatment [154,155,156]. However, the most effective dietary pattern for its management has not been established, and those dietary interventions are similar regarding high fiber intake, vitamins, minerals, and polyphenols. The potential prebiotic effects of these dietary interventions have yet to be thoroughly studied. The number of prebiotics must be considered an essential ingredient for optimal results in managing metabolic diseases. For example, the quantity of prebiotics can vary for those with the same fiber content. Furthermore, a more direct glycemic manipulation is a low-carbohydrate diet [157]. The short-term effects (less than 6 months) of a low-carbohydrate diet are efficacious for the reduction in fat mass and remission of type 2 diabetes; however, there is a diminishment of weight loss and metabolic cardiovascular benefits beyond 6 months [157]. A long-term effect of the low-carbohydrate diet was associated with a higher risk of all-cause mortality based on 272,216 individuals in observational studies and large-scale trials [158]. This result may be due to an increased mortality rate when carbohydrates were exchanged for animal-derived fat or protein, but there was decreased mortality when they were exchanged with plant-based sources [159]. This outcome also confirmed the importance of the quality of carbohydrate intake. Still, these diets are hardly quantified and may cause malnutrition [160]. The SIM diet may encompass its immunomodulatory effects, biological impacts, and underlying molecular mechanisms [161] to solve this problem.

## 14. Adverse Effects on Gastrointestinal Symptoms and Tolerance

Doubtless, there are two sides to every coin. As prebiotics and potential prebiotics interact with the gut microbiome, luminal gas production and osmotic effects caused by fermentation in the gut may cause more gastrointestinal symptoms, such as diarrhea, bloating, borborygmi, belching, and nausea or vomiting. Notably, NSPs and resistant starch cause few GI symptoms [162], whereas shorter chains of carbohydrates cause more [163]. A meta-analysis of findings from 103 clinical trials in adults without gastrointestinal disease who reported gastrointestinal effects, including those involving tolerance (e.g., bloating, flatulence, and borborygmi/rumbling) and function (e.g., transit time, stool frequency, and stool consistency), suggested that different types of NSPs and fermentable carbohydrates should have tolerable intake dose recommendations for their consumption [164]. Moreover, depending on their psychological conditions, individuals may have different levels of gastrointestinal tolerance [165]. Dietitians and health professionals must proactively inquire about the increased intake of natural prebiotics for individuals with metabolic diseases. This proactive approach can help reduce the chances that patients will discontinue their medication when they experience common gastrointestinal symptoms with suitable foods. It will also potentially improve patients’ overall health and well-being. Furthermore, the gut microbiome affects virtually all metabolic diseases. However, the understanding of mechanisms of dietary manipulation, including the role of the microbiota and its metabolites and other chemical components, varies considerably from one benefit area to the other. This review helps us better understand the mechanisms of new natural food therapy in addressing the SIM and how it interacts with the gut–organ axis. The findings call for further coordinated state-of-the-art clinical research to elucidate the mechanisms of interaction between food and the body (host) to document the gut microbiome effects on metabolic diseases.

## 15. Conclusions

Understanding the role of natural prebiotics in managing metabolic diseases is a key area for future research. This holistic SIM approach to dietary recommendations can potentially yield more tailored and effective dietary approaches for individuals with metabolic diseases. By delving into underlying mechanisms and therapeutic potential, we can identify more targeted and personalized approaches to metabolic disease management. However, further well-designed large-scale interventional studies are urgently needed to ensure the generalizability of these findings and to fully understand the potential of natural prebiotics in managing metabolic diseases.

## Figures and Tables

**Figure 1 biology-13-00498-f001:**
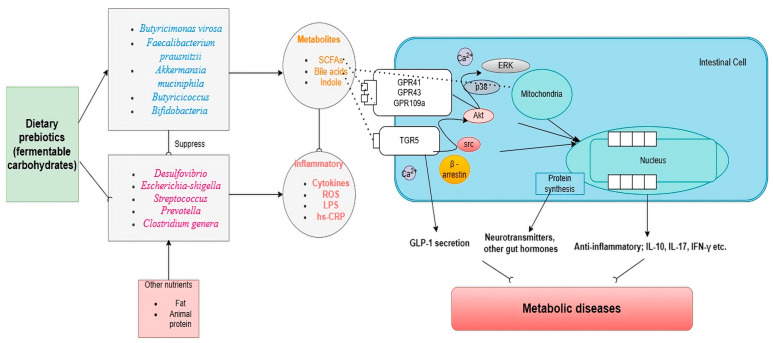
Dietary prebiotics reshape the gut microbiome and restore metabolic diseases. Abbreviations: TGR5: thiol guanosine receptor-5, GPR: G protein-coupled receptor, LPS: lipopolysaccharide, SCFA: short-chain fatty acid, Hs-CRP: high sensitivity C-reactive protein; ROS: reactive oxygen species; Akt: protein kinase B; p38; mitogen-activated protein kinases; ERK: extracellular signal-regulated kinase 1. → indicates induction and ⸺⸦ indicates suppression.

**Figure 2 biology-13-00498-f002:**
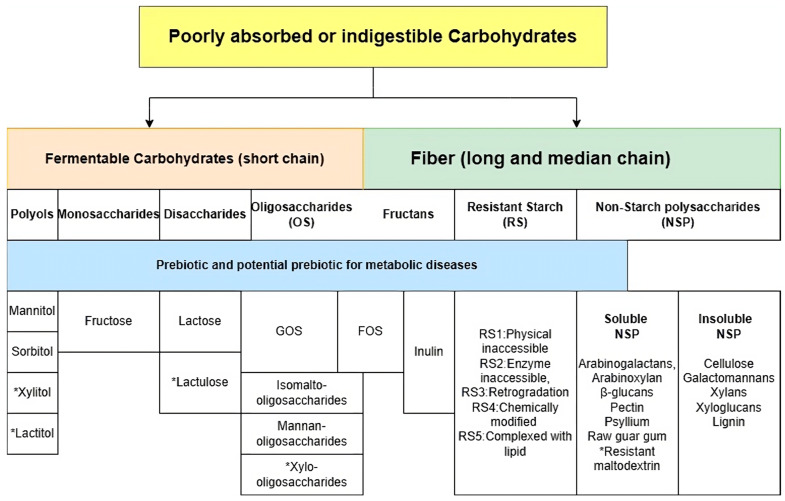
Categories of fermentable carbohydrates. Abbreviation: GOS: galacto-oligosaccharides, FOS: fructo-oligosaccharides. * Artificial compounds.

**Table 1 biology-13-00498-t001:** Category of high fermentable carbohydrate content in natural foods and alteration of the microbiome. *↓* indicates the reduction of the relative abundance of the gut microbiome.

Soluble NSP	Oligosaccharides	Resistant Starch and Inulin	Di- and Mono-Saccharides	Polyols
Oat bran	Wheat	Lentils	Dairy	Mushrooms
Barley	Pulses	Oats	Some Fruits (Apples, pears, etc.)	Cherry
Seeds	Figs	Barley		Dates
Apples	Garlic	Banana		
Oranges	Onion			
Carrots	Nuts			
Genus [70]	Genus [77,80]	Genus [84,85,87]	Family [99]	Family [117]
*Bacteroides*	*Akkermansia*	*Akkermansia*	Bacteroidaceae	Verrucomicrobia
*Roseburia*	*Barnesiella*	*Ruminococcus*	Lactobacillaceae	Phylum [117]
*Coprobacillus*	*Odoribacter*	*Victivallis*	Enterococcaceae	*↓* Cyanobacteria
Species [69]	*Coprococcus*	*Comamonas*	Streptococcaceae	Genus [117]
*Lachnospira eligens*	*Butyricicoccus*	*Lactobacillus*	Genus [113,114]	*Bacteroides*
*Faecalibacterium prausnitzii*	*Bifidobacterium*	*Bifidobacterium*	*Anaerostipes*	*Phascolarctobacterium*
		*Turicibacter*	*Coprococcus*	*↓ Escherichia-Shigella*
		*Phascolarctobacterium*	*Ruminococcus*	
		*Blautia*	*Erysipelatoclostridium*	
		*↓ Desulfovibrio*

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
