# Peer review of "The Therapeutic Potential of the Specific Intestinal Microbiome (SIM) Diet on Metabolic Diseases"

_biology, 2024, doi:10.3390/biology13070498_

Round 1

Reviewer 1 Report

Comments and Suggestions for Authors

This review provides interesting information and facts about the role of the gut microbiome and its metabolites in different diseases and at different levels. This review highlights the importance of diet-mediated interventions compared to other types of interventions for the betterment of various health conditions associated with diet and gut microbiota. The authors are requested to thoroughly check the review for its language correctness. Authors are suggested to consider the following comments and suggestions to further improve the quality of this review. 

1.        However, 55 apart from the dietary fibre, other short-chain or medium-chain carbohydrates also induce 56 specific intestinal microbiome (SIM) and their metabolites, such as secondary bile acids 57 and short-chain fatty acids (SCFA),

In the above text, provide examples for short-chain or medium-chain carbohydrates with references.

2.        Provide explanation about the term “gut microbiota assistance” in the line 65.

3.        However, cur- 65 rent knowledge in the short and medium chain of carbohydrates has limited studies on 66 the pattern of intestinal microbiota in individuals with metabolic diseases[6].

In the above text rephrase the text to explain clearly about the state of present knowledge about short and medium chain of carbohydrates.

4.        Different strains have its suitable environments for colonization, and foods are essential for its development[16].

Rephrase the above text like this “Different strains have suitable environments for colonization, and foods are essential for their development”.

5.        and microbiota are predominated by carbohydrate polymers like hu- 114 mans[17].

In the text above, do the authors mean that the composition of the microbiota is predominated by carbohydrate polymers like humans? Rephrase the text for proper understandability.

6.        Cur- 169 rent animal study has uncovered the unique function of Butyricimonas virosa, which not 170 only belonged to a SCFAs-producing bacteria but also exhibited its novel function in glu- 171 cose regulation that was linked to the upregulation of GLP-1 receptors rather than related 172 to its production of SCFAs[37].

Rephrase the above text like this for proper understanding “A current animal study has uncovered the unique function of Butyricimonas virosa, which not only belonged to a SCFAs-producing bacteria but also exhibited a novel function in glucose regulation that was linked to the upregulation of GLP-1 receptors rather than its production of SCFAs[37].”

7.        SCFA-mediated HDAC inhibition promotes chroma- 180 tin acetylation and downstream consequences of cardiovascular and microvascular met- 181 abolic diseases, offering a promising avenue for future research and potential therapeutic 182 interventions.

In the text above, do the authors imply that SCFA-mediated HDAC inhibition results in cardiovascular complications? Please rephrase the text to make it clearer for readers.

8.        While inulin has shown potential benefits in managing diabetes mellitus and obesity[79], 309 it is important to note that its effects of inulin can vary in among individuals and contexts.

Correct the grammar of the above text.

9.        Correct the grammar of this text “However, lactulose belongs to artificial sugars, and we had been discussed 356 it as a prebiotic in the previous review[93]. ” like this “However, lactulose belongs to artificial sugars, and we discussed 356 it as a prebiotic in the previous review.

10.  For table 2, provide more comprehensive description of the contents and adjust the format of the table so that it is not distorted as it is now.

11.  In the last part of the discussion, authors should also address the gaps in knowledge or research on the role of the microbiota and its metabolites and other components in metabolic profiles.

Comments on the Quality of English Language

The authors should review this work for language correctness and rephrase the sentences that are not so clear. 

Author Response

Reviewer 1

This review provides interesting information and facts about the role of the gut microbiome and its metabolites in different diseases and at different levels. This review highlights the importance of diet-mediated interventions compared to other types of interventions for the betterment of various health conditions associated with diet and gut microbiota. The authors are requested to thoroughly check the review for its language correctness. Authors are suggested to consider the following comments and suggestions to further improve the quality of this review. 

  1. However, 55 apart from the dietary fibre, other short-chain or medium-chain carbohydrates also induce 56 specific intestinal microbiome (SIM) and their metabolites, such as secondary bile acids 57 and short-chain fatty acids (SCFA),

In the above text, provide examples for short-chain or medium-chain carbohydrates with references.

Thank you for your suggestion. We have revised the sentence.

  1. Provide explanation about the term “gut microbiota assistance” in the line 65.

Thank you for your suggestion. We added, “in a range of bodily functions, including protection of pathogens and regulation of immune functions.”

  1. However, cur- 65 rent knowledge in the short and medium chain of carbohydrates has limited studies on 66 the pattern of intestinal microbiota in individuals with metabolic diseases[6].

In the above text rephrase the text to explain clearly about the state of present knowledge about short and medium chain of carbohydrates.

 Thank you for your suggestion. We have rephrased the sentence, “The effects of macronutrient metabolism by the gut microbiome on human health have been extensively studied, research on the impact of short and medium chain carbohydrates remains limited.”

  1. Different strains have its suitable environments for colonization, and foods are essential for its development[16].

Rephrase the above text like this “Different strains have suitable environments for colonization, and foods are essential for their development”.

Thank you for your suggestion. We have rephrased the sentence “The human gut microbiome is a complex and dynamic ecosystem, in which different strains of microorganisms compete for different resources to effectively colonise it.”

  1. and microbiota are predominated by carbohydrate polymers like hu- 114 mans[17].

In the text above, do the authors mean that the composition of the microbiota is predominated by carbohydrate polymers like humans? Rephrase the text for proper understandability.

Thank you for your suggestion. We have rephrased the sentence, “the composition of the microbiota is predominated by carbohydrate polymers.”

  1. Cur- 169 rent animal study has uncovered the unique function of Butyricimonas virosa, which not 170 only belonged to a SCFAs-producing bacteria but also exhibited its novel function in glu- 171 cose regulation that was linked to the upregulation of GLP-1 receptors rather than related 172 to its production of SCFAs[37].

Rephrase the above text like this for proper understanding “A current animal study has uncovered the unique function of Butyricimonas virosa, which not only belonged to a SCFAs-producing bacteria but also exhibited a novel function in glucose regulation that was linked to the upregulation of GLP-1 receptors rather than its production of SCFAs[37].”

Thank you for your suggestion. We have rephrased the sentence.

  1. SCFA-mediated HDAC inhibition promotes chroma- 180 tin acetylation and downstream consequences of cardiovascular and microvascular met- 181 abolic diseases, offering a promising avenue for future research and potential therapeutic 182 interventions.

In the text above, do the authors imply that SCFA-mediated HDAC inhibition results in cardiovascular complications? Please rephrase the text to make it clearer for readers.

 Thank you for your suggestion. We have rephrased the sentence.

  1. While inulin has shown potential benefits in managing diabetes mellitus and obesity[79], 309 it is important to note that its effects of inulin can vary in among individuals and contexts.

Correct the grammar of the above text.

Thank you for your suggestion. We have revised the sentence as “it is important to note that the effects of inulin can vary among individuals”

  1. Correct the grammar of this text “However, lactulose belongs to artificial sugars, and we had been discussed 356 it as a prebiotic in the previous review[93]. ” like this “However, lactulose belongs to artificial sugars, and we discussed 356 it as a prebiotic in the previous review.

Thank you for your suggestion. We have rephrased the sentence.

  1. For table 2, provide more comprehensive description of the contents and adjust the format of the table so that it is not distorted as it is now.

 Thank you for your suggestion. We have revised the description of the contents.

  1. In the last part of the discussion, authors should also address the gaps in knowledge or research on the role of the microbiota and its metabolites and other components in metabolic profiles.

Thank you for your suggestion. We added, “Furthermore, the gut microbiome affects virtually all metabolic diseases. However, the understanding of mechanisms of dietary manipulation, including the role of the microbiota and its metabolites and other chemical components, varies considerably from one benefit area to the other. This review helps us better understand the mechanisms of new natural food therapy in addressing SIM and how it interacts with the gut-organ axis. The findings call for further coordinated state-of-the-art clinical research to elucidate the mechanisms of interaction between food and the body (host) to document the gut microbiome effects on metabolic diseases.”

Reviewer 2 Report

Comments and Suggestions for Authors

Dear authors,

Thank You for the opportunity to review the manuscript titled: "Intestinal Crosstalk between dietary prebiotics and Specific Intestinal Microbiome (SIM) for the treatment of metabolic diseases" for Biology.

The manuscript in question provides an interesting insight into the roles and importance of probiotics in today’s diet and health promotion.

Overall, this was a pretty complicated read, mainly due to language barriers, but I believe it might be an interesting read for many other policy professionals involved with public health, nutrition and wellbeing if major language editing takes place…

I have some suggestions for the respectable authors that mainly stem from misunderstandings due to improper language use.

Lines 55-59:

There is something wrong with this sentence, specifically in part “…which have been examined the importance in the role…” Please, rephrase.

Line 61:

“feeding with natural prebiotics”? Food maybe? Please, rephrase.

Lines 100-104:

“In the previous review, we summarized the characteristics of gut microbiota in newly diagnosed diabetes and these microbiota pattern can be restored with antidiabetic treatment including the Western and Chinese medicine [13]. Due to the impact of drugs on reshaping the gut microbiome, probiotic administration is another option for microbiome manipulation”

You have shown that disease treatment can affect SIM patterns. I don’t see hoe the second sentence is linked to that finding.

Lines 114-115

“microbiota are predominated by carbohydrate polymers like humans”

What does this mean?

Lines 139-140

“A study involving a blinded, randomized, cross-over dietary intervention on healthy individuals further solidified these findings.”

What is the reference to that study? Number 9? If so, it should be mentioned in this sentence.

Lines 163-167

“Additionally, SCFAs induce enteroendocrine L cells that control appetite and upregulate intestinal gluconeogenesis gene expression that controls glucose regulation in support of the Brown et al. and Ørgaard A. et al. studies that significantly inhibited the lipolysis of primary human fat cells and stimulates the secretion of glucagon-like peptide-1 (GLP-1)  [34] and somatostatin hormone [35].”

Please, rephrase this…

Line 184

“microbiota-medicated metabolite”

Mediated?

Lines 226-229

“For example, a meta-analysis by He and colleagues did not observe beta-glucans associations with HbA1c, fasting glucose, and fasting insulin of T2D, hyperlipidaemic and overweight, but oligosaccharides in oats did[49].”

It seems that oligosaccharides from oats did observe those things that He and his coauthors did not. Please, rephrase.

Lines 267-269

“More critical in psyllium administration was discussed in its considerable cholesterol-lowering effects.”

Please, rephrase.

Lines 271- 287

Subsection Oligosaccharides

There is no need to capitalize names of specific oligosaccharides.

In addition, “…Mannan, which is converted by Galactomannan polysaccharides (soluble fibre) during heating…”

How can mannan be converted into anything by polysaccharides?

Line 423

e.g. “…guar gum… ameliorates waist circumference…”

These language mistakes make this manuscript very hard to read through…

Overall, English language needs to be edited. Some parts of the manuscript don’t flow naturally and are hard to follow (e.g. SIM and metabolic diseases and several others).

I truly believe that the authors are on to an interesting and important topic, but are limited by a language barrier, and that this manuscript world benefit greatly by language editing.

Sincerely,

The reviewer

Comments on the Quality of English Language

Comments are given in the "Comments and Suggestions for Authors
" section.

Author Response

Reviewer 2

The manuscript in question provides an interesting insight into the roles and importance of probiotics in today’s diet and health promotion.

Overall, this was a pretty complicated read, mainly due to language barriers, but I believe it might be an interesting read for many other policy professionals involved with public health, nutrition and wellbeing if major language editing takes place…

I have some suggestions for the respectable authors that mainly stem from misunderstandings due to improper language use.

Lines 55-59:

There is something wrong with this sentence, specifically in part “…which have been examined the importance in the role…” Please, rephrase.

Thank you for your suggestion. We have rephrased the sentence.

Line 61:

“feeding with natural prebiotics”? Food maybe? Please, rephrase.

Thank you for your suggestion. We have rephrased the sentence.

 “Bioactive metabolites that originate from gut microbiota are known to be produced through consuming natural food.”

Lines 100-104:

“In the previous review, we summarized the characteristics of gut microbiota in newly diagnosed diabetes and these microbiota pattern can be restored with antidiabetic treatment including the Western and Chinese medicine [13]. Due to the impact of drugs on reshaping the gut microbiome, probiotic administration is another option for microbiome manipulation”

You have shown that disease treatment can affect SIM patterns. I don’t see hoe the second sentence is linked to that finding.

Thank you for your suggestion. We changed it to “Considering the significant impact of drugs on reshaping the gut microbiome, the administration of probiotics stands out as a direct and effective method for microbiome manipulation.

Lines 114-115

“microbiota are predominated by carbohydrate polymers like humans”

What does this mean?

Thank you for your suggestion. We changed it to “the composition of the microbiota is predominated by carbohydrate polymers”

Lines 139-140

“A study involving a blinded, randomized, cross-over dietary intervention on healthy individuals further solidified these findings.”

What is the reference to that study? Number 9? If so, it should be mentioned in this sentence.

Thank you for your suggestion. We have added the reference.

Lines 163-167

“Additionally, SCFAs induce enteroendocrine L cells that control appetite and upregulate intestinal gluconeogenesis gene expression that controls glucose regulation in support of the Brown et al. and Ørgaard A. et al. studies that significantly inhibited the lipolysis of primary human fat cells and stimulates the secretion of glucagon-like peptide-1 (GLP-1)  [34] and somatostatin hormone [35].”

Please, rephrase this…

Thank you for your suggestion. We changed it to “Additionally, the observations are in line with the studies by Brown et al. and Ørgaard A. et al. which demonstrated significant inhibition of lipolysis in primary human fat cells and stimulation of glucagon-like peptide-1 (GLP-1) and somatostatin hormones secretion.”

Line 184

“microbiota-medicated metabolite”

Mediated?

Thank you for your correction.

Lines 226-229

“For example, a meta-analysis by He and colleagues did not observe beta-glucans associations with HbA1c, fasting glucose, and fasting insulin of T2D, hyperlipidaemic and overweight, but oligosaccharides in oats did[49].”

It seems that oligosaccharides from oats did observe those things that He and his coauthors did not. Please, rephrase.

Thank you for your suggestion. We have changed it to “For example, a meta-analysis by He and colleagues did not find associations between beta-glucans and HbA1c, fasting glucose, and fasting insulin in overweight individuals or those with T2D or hyperlipidemia. However, the study did find associations between oats (which are high in oligosaccharides) and these variables.”

Lines 267-269

“More critical in psyllium administration was discussed in its considerable cholesterol-lowering effects.”

Please, rephrase.

Thank you for your suggestion. We have changed it to “More importantly, the significant cholesterol-lowering effects of psyllium were highly discussed.”

Lines 271- 287

Subsection Oligosaccharides

There is no need to capitalize names of specific oligosaccharides.

Thank you for your suggestion.

In addition, “…Mannan, which is converted by Galactomannan polysaccharides (soluble fibre) during heating…”

How can mannan be converted into anything by polysaccharides?

Thank you for your suggestion. We added the “involvement of hydration and swelling, disruption of hydrogen bonds, gel formation and increased viscosity in industrial processes” into the sentence.

Line 423

e.g. “…guar gum… ameliorates waist circumference…”

These language mistakes make this manuscript very hard to read through…

Thank you for your suggestions. We have changed to “An example of this is when individuals with diabetes add soluble fibre guar gum to their regular diet, which leads to improvements in waist circumference, HbA1c, 24-hour urinary albumin excretion, and serum trans-fatty acids compared to their baseline levels. This also results in greater weight reduction compared to a control group.”

Overall, English language needs to be edited. Some parts of the manuscript don’t flow naturally and are hard to follow (e.g. SIM and metabolic diseases and several others).

Thank you so much for your suggestion, we have undergone the English edits.

I truly believe that the authors are on to an interesting and important topic, but are limited by a language barrier, and that this manuscript world benefit greatly by language editing.

Reviewer 3 Report

Comments and Suggestions for Authors

The topic of the review article is good, but some notes should be taken into account:

Line 20, please replace the word “carbohydrate” with the term “prebiotic.” Not all carbohydrates act as prebiotics.

In general, some points were mentioned at length, while essential points were mentioned briefly. For example, Figure 1 the mechanism of dietary prebiotics involving the crosstalk of gut microbiome and its metabolites on metabolic diseases. A fundamental point in the article that was not mentioned except in the figure and its title.

“Microbiome and Immunometabolism”   It would be good if you replaced this title with another title dealing with the metabolic products of the gut microbiota.

With the exception of mentioning the sugar lactulose, it is not appropriate to mention disaccharides and monosaccharides in the context of discussing prebiotics and the gut mycobiota, as they are not prebiotics. Free fructose commonly causes metabolic syndrome and should not be confused with fruit fructose bound within fruits.

The term non-alcoholic fatty liver disease is abbreviated as NAFLD, not MAFLD (line 436).

Table 1 is not homogeneous and needs more clarification, as well as an explanation in some detail in the text.

“Comparison between current dietary therapies on metabolic diseases”, The paragraph under this title mentioned the effect of different dietary patterns on metabolic problems. I could not understand the link between that and prebiotics and their effect on the gut microbiota. Please clarify.

Comments on the Quality of English Language

Minor editing of the English language is required.

Author Response

Reviewer 3

l Line 20, please replace the word “carbohydrate” with the term “prebiotic.” Not all carbohydrates act as prebiotics.

Thank you for your suggestion.

l In general, some points were mentioned at length, while essential points were mentioned briefly. For example, Figure 1 the mechanism of dietary prebiotics involving the crosstalk of gut microbiome and its metabolites on metabolic diseases. A fundamental point in the article that was not mentioned except in the figure and its title.

Thank you for your suggestion. We should change the renamed the subtitle of “Microbiome and Immunometabolism” to the crosstalk of gut microbiome and their metabolites as this session mainly discusses the microbiome and metabolites on metabolism.

l “Microbiome and Immunometabolism”   It would be good if you replaced this title with another title dealing with the metabolic products of the gut microbiota.

Thank you for your suggestion. We changed to “Crosstalk of gut microbiome and its metabolites”

l With the exception of mentioning the sugar lactulose, it is not appropriate to mention disaccharides and monosaccharides in the context of discussing prebiotics and the gut mycobiota, as they are not prebiotics. Free fructose commonly causes metabolic syndrome and should not be confused with fruit fructose bound within fruits.

Thank you for your suggestion. This involves the concept of fructose absorption in humans. Fruits and vegetables contain fructose and glucose, but when fructose is higher than glucose in certain foods, fructose is not fully absorbed by the small intestine, leaves the large intestine, and is fermented by the gut microbiome. Therefore, we should indicate “excess fructose” rather than fructose alone.

Gibson PR. History of the low FODMAP diet. J Gastroenterol Hepatol. 2017 Mar;32 Suppl 1:5-7. doi: 10.1111/jgh.13685. PMID: 28244673.

l The term non-alcoholic fatty liver disease is abbreviated as NAFLD, not MAFLD (line 436).

Thank you for your correction.

l Table 1 is not homogeneous and needs more clarification, as well as an explanation in some detail in the text.

Thank you for your suggestion. We added more information on it.

l “Comparison between current dietary therapies on metabolic diseases”, The paragraph under this title mentioned the effect of different dietary patterns on metabolic problems. I could not understand the link between that and prebiotics and their effect on the gut microbiota. Please clarify.

Thank you so much for raising the point. We added the sentence, “The potential prebiotic effects of these dietary interventions have yet to be thoroughly studied. The number of prebiotics must be considered an essential ingredient for optimal results in managing metabolic diseases. For example, the quantity of prebiotics can vary for those with the same fibre content.”

Reviewer 4 Report

Comments and Suggestions for Authors

The peer-reviewed literature review raises a very important and already quite well studied problem of the relationship between the human macroorganism, the components of its diet (prebiotics) and the bacterial population of its intestines. Dozens of literature reviews on this topic and thousands of experimental studies have been published. Probably, the next literature review should be more in-depth and should provide an original look at the already accumulated data.

Unfortunately, I did not see any scientific novelty in the article being reviewed. The authors summarized more than 150 literary sources, but the degree of their understanding of the material is far from the modern scientific level. It is likely that the Wikipedia article presents the problem of the interaction of the human gut microbiome with prebiotics more systematically (https://en.wikipedia.org/wiki/Gut_microbiota) than the peer-reviewed article.

Over the past 5 years, dozens of literature reviews on this topic have been published, for example, look for the phrase “dietary prebiotics and microbiome” on the website https://www.ncbi.nlm.nih.gov/pmc. These literature reviews analyze changes in the abundance of bacterial taxonomic groups at a higher level (for example, see the figures in this study: https://www.ncbi.nlm.nih.gov/pmc/articles/PMC9336045/ ). Authors should rely on the best examples of such literature reviews. Now the article under review partially contains generalized information, but it contains dozens of fragments of superficial judgments that do not correspond to the modern understanding of the object of study.

The article needs detailed analysis by microbiologists on the one hand (in my opinion, this is the weakest aspect of the peer-reviewed literature review) and organic chemists on the other hand (authors often confuse chemical terms). The interaction of microflora with organic substances occurs in conditions of various metabolic disorders (enzymatic disorders in the liver, endocrine, immune disorders) and intestinal digestive processes under the influence of bile and pancreatic enzymes. These pathological changes cause new changes in metabolic processes in the macroorganism. Unfortunately, I did not see a holistic understanding of all these complex interactions in the peer-reviewed literature review.

Technical shortcomings of the manuscript

1. Simple Summary should be expanded to 4-5 sentences, understandable for journalists and non-specialists. Lines 20-25 will probably be more understandable to non-specialists, so it is better to add them to the Simple Summary.

2. Abstract should be expanded to 15-18 lines. In the Abstract text, the most important substances and Latin names of microorganisms from the text of the article should be mentioned. This will improve the visibility of the article in Web of Science and Scopus and increase the number of links to it.

3. Line 36: The image does not contain any new information, even for a layman. Probably, you need to add 30-35 names of substances and microorganisms to this image, connecting them with arrows according to the meaning of the article.

4. It is probably necessary to indicate the names of bacterial genera more carefully in Table 1. It would probably be more correct, after the birth of bacteria, to indicate with one, two or three arrows the degree of stimulation of the reproduction of representatives of each group of microorganisms. In the note under this table you need to indicate the sources of literature that confirm the data in the table. The table text contains a large number of errors, typos, and incorrect italics. The table as a whole needs to be made more scientific. The authors should study more deeply the ecology of these groups of bacteria.

5. In the text of the article, only generic and species names of bacteria should be italicized. Families, phyla and other taxonomic Latin names cannot be italicized. "spp." Italics cannot be used (for example, line 92).

6. The numbers after the generic names of bacteria should be removed (for example, lines 299, 317 and in Table 1).

7. There must be a space before the reference to the literature, for example line 295, 297, 302.

8. There must be a space between the number and the unit of measurement, for example line 322.

9. Only proper names should be written with a capital letter. Many terms need to be capitalized, for example line 273, Figure 2, Table 1.

10. The word “shigella” must be capitalized throughout the text of the article.

11. The literature is not formatted according to the requirements of the journal.

12. A significant part of the literature is devoid of indexes. You need to add them.

Author Response

Reviewer 4

The peer-reviewed literature review raises a very important and already quite well studied problem of the relationship between the human macroorganism, the components of its diet (prebiotics) and the bacterial population of its intestines. Dozens of literature reviews on this topic and thousands of experimental studies have been published. Probably, the next literature review should be more in-depth and should provide an original look at the already accumulated data.

Unfortunately, I did not see any scientific novelty in the article being reviewed. The authors summarized more than 150 literary sources, but the degree of their understanding of the material is far from the modern scientific level. It is likely that the Wikipedia article presents the problem of the interaction of the human gut microbiome with prebiotics more systematically (https://en.wikipedia.org/wiki/Gut_microbiota) than the peer-reviewed article.

Over the past 5 years, dozens of literature reviews on this topic have been published, for example, look for the phrase “dietary prebiotics and microbiome” on the website https://www.ncbi.nlm.nih.gov/pmc. These literature reviews analyze changes in the abundance of bacterial taxonomic groups at a higher level (for example, see the figures in this study: https://www.ncbi.nlm.nih.gov/pmc/articles/PMC9336045/ ). Authors should rely on the best examples of such literature reviews. Now the article under review partially contains generalized information, but it contains dozens of fragments of superficial judgments that do not correspond to the modern understanding of the object of study.

The article needs detailed analysis by microbiologists on the one hand (in my opinion, this is the weakest aspect of the peer-reviewed literature review) and organic chemists on the other hand (authors often confuse chemical terms). The interaction of microflora with organic substances occurs in conditions of various metabolic disorders (enzymatic disorders in the liver, endocrine, immune disorders) and intestinal digestive processes under the influence of bile and pancreatic enzymes. These pathological changes cause new changes in metabolic processes in the macroorganism. Unfortunately, I did not see a holistic understanding of all these complex interactions in the peer-reviewed literature review.

Thank you for your suggestion. We agreed that the holistic unmet understanding in crosstalk with microorganism and their metabolites on metabolic diseases. In this review, we would like to suggest natural foods containing different types of prebiotics. Some of them are not classified as prebiotics but work with the gut microbiome in managing metabolic diseases. This new angle in dietary manipulation holds great promise, potentially revolutionizing the efficiency of dietary intervention. This exciting prospect offers a new way of managing metabolic diseases, as most of the current dietary interventions only emphasize fibre intake. For example, the quantity of prebiotics can vary for those with the same amount of fibre. This knowledge forms a robust basis for creating a new natural therapy to quantify the amount of prebiotic intakes that current dietary interventions do not mention.

Technical shortcomings of the manuscript

  1. Simple Summary should be expanded to 4-5 sentences, understandable for journalists and non-specialists. Lines 20-25 will probably be more understandable to non-specialists, so it is better to add them to the Simple Summary.

Thank you for your suggestion. We have changed the Lines 20-25 to the simple summary.

  1. Abstract should be expanded to 15-18 lines. In the Abstract text, the most important substances and Latin names of microorganisms from the text of the article should be mentioned. This will improve the visibility of the article in Web of Science and Scopus and increase the number of links to it.

Thank you for your suggestion. We added, “These metabolic-related microorganisms include Butyricicoccus, Akkermansia, and Phascolarctobacterium.”

  1. Line 36: The image does not contain any new information, even for a layman. Probably, you need to add 30-35 names of substances and microorganisms to this image, connecting them with arrows according to the meaning of the article.

Thank you for your suggestion. This is a graphical abstract that should be kept simple to be eye-catching, but Figure 2 provides more information about the SIM diet and it interacts with metabolites.

  1. It is probably necessary to indicate the names of bacterial genera more carefully in Table 1. It would probably be more correct, after the birth of bacteria, to indicate with one, two or three arrows the degree of stimulation of the reproduction of representatives of each group of microorganisms. In the note under this table you need to indicate the sources of literature that confirm the data in the table. The table text contains a large number of errors, typos, and incorrect italics. The table as a whole needs to be made more scientific. The authors should study more deeply the ecology of these groups of bacteria.

Thank you for your suggestion. We have corrected the table with taxonomy and references.

  1. In the text of the article, only generic and species names of bacteria should be italicized. Families, phyla and other taxonomic Latin names cannot be italicized. "spp." Italics cannot be used (for example, line 92).

Thank you for your suggestion. We have amended it in the manuscript.  

  1. The numbers after the generic names of bacteria should be removed (for example, lines 299, 317 and in Table 1).

Thank you for your correction.

  1. There must be a space before the reference to the literature, for example line 295, 297, 302.

Thank you for your suggestion. We have amended it in the manuscript.  

  1. There must be a space between the number and the unit of measurement, for example line 322.

Thank you for your suggestion. We have amended it in the manuscript.  

  1. Only proper names should be written with a capital letter. Many terms need to be capitalized, for example line 273, Figure 2, Table 1.

Thank you for your suggestion.

  1. The word “shigella” must be capitalized throughout the text of the article.

Thank you for your correction.

  1. The literature is not formatted according to the requirements of the journal.

Thank you for your suggestion; we have amended the table as required.

  1. A significant part of the literature is devoid of indexes. You need to add them.

Thank you for your suggestion. We have added all of them.

Round 2

Reviewer 1 Report

Comments and Suggestions for Authors

The changes made to the review are satisfactory and the authors have addressed all of my comments and suggestions. 

Author Response

Thank you for giving us the opportunity to address the unmet needs of current dietary therapy by modifying the gut microbiome. 

Reviewer 3 Report

Comments and Suggestions for Authors

Great effort has been made by the authors to improve the manuscript.

Author Response

Thank you so much for giving us valuable comments for improving this review.  

Reviewer 4 Report

Comments and Suggestions for Authors

The authors did not radically improve the article. Technical deficiencies have been addressed, but the quality of the article does not match the best examples of literature reviews on this topic.

Author Response

Thank you so much for giving us valuable comments for improving this review, and we changed the title to " The Therapeutic Potential of the Specific Intestinal Microbiome (SIM) Diet for the treatment of Metabolic Diseases" because we agreed that the core of this review is discussing the potential use of dietary prebiotics on metabolic diseases via changes in the microbiome and its metabolites rather than largely discuss the interactions between dietary prebiotics and SIM. 

We further include the mechanisms of microbial metabolites that support changes in gut hormones, brown adipose tissue activation, insulin sensitivity and resistance via the gut-brain axis of specific SCFAs and conjugated bile acids in deep:

SCFAs were shown to decrease the serum concentration of ghrelin (appetite hormone) and stimulate the production of GLP-1 and Peptide YY by activating GPR43 and GPR41. This involved the mechanisms in the gut-brain neural circuit; an animal study observed that butyrate administration suppressed the activity of orexigenic neurons and decreased neuronal activity in the brainstem. Therefore, this reduced the appetite and activated brown adipose tissue by utilising plasma triglyceride-derived fatty acids. More specifically, in one of SCFAs acetate, a study using labelled carbohydrate and positron emission tomography (PET)-scanning to investigate the SCFAs and its effect on the host’s appetite, intraperitoneal acetate is fermented by the gut microbiome and taken by the brain, and this resulted in appetite suppression and hypothalamic neuronal activation patterning.

For instance, a water-soluble bile acid, tauroursodeoxycholic acid (TUDCA), is produced by the conjugation of UDCA with taurine in the body. This compound has shown a 30 % improvement in hepatic and muscle insulin sensitivity in people with obesity compared to the placebo. Another promising interaction of conjugated secondary bile acid is glycodeoxycholic acid (GDCA), which forms with deoxycholic acid and another amino acid, glycine, in the liver. GDCA was correlated with reduced insulin resistance via the communication with gut microbiome and interleukin-22 secretion.